# Expanding and Enhancing Food and Nutrition Education in New York City Public Schools: An Examination of Program Characteristics and Distribution

**DOI:** 10.3390/nu12082423

**Published:** 2020-08-12

**Authors:** Pamela Koch, Julia McCarthy, Claire Raffel, Heewon L. Gray, Laura A. Guerra

**Affiliations:** 1Department of Health and Behavior Studies, Teachers College, Columbia University; New York, NY 10027, USA; pak14@tc.columbia.edu (P.K.); cu2155@tc.columbia.edu (C.R.); lag2177@tc.columbia.edu (L.A.G.); 2College of Public Health, University of South Florida; Tampa, FL 33620, USA; hlgray@usf.edu

**Keywords:** food and nutrition education, schools, practice, school and community partnerships

## Abstract

To expand their capacity, many schools partner with food and nutrition education programs (FNPs). Public policies and funding can support FNPs, but comprehensive data on the organizations that run FNPs, their program characteristics, or distribution across schools did not exist in NYC. This study aims to help local education and health agencies assess the characteristics of food and nutrition education in schools, as well as to measure progress implementing school policies and practices. A cross-sectional study on NYC FNPs was conducted during the 2016–2017 school year. Survey data on organizations and the FNPs they operate were collected. Data on schools in which FNPs operate were gathered. To determine distribution of FNPs across schools and by school demographics, the database of FNPs in schools was combined with a publicly available database of NYC schools. In 2016–2017, 40 organizations operated 101 FNPs in 56% of NYC public schools. These FNPs varied by goals, content, activities, location, and populations served. Information on these variations can help policymakers, advocates, funders, and schools expand school-based food and nutrition education. To ensure equitable access, more coordination, investment, and collaboration are needed.

## 1. Introduction

Food and nutrition education engages students in hands-on activities, combining direct education with environmental reinforcements at the individual, community, and policy levels to build motivation, skills, and knowledge to make healthy choices [1,2,3]. Food and nutrition education not only promotes healthy behaviors and body weight, but it can also improve students′ academic performance and participation in school meals [4,5,6,7,8].

Research recommends that students partake in 30 to 50 h of behaviorally focused, high quality food and nutrition education each year [9]. However, academic requirements, standardized testing, and staff expertise can limit schools′ capacity to provide the recommended amount [10]. According to the U.S. Department of Agriculture′s 2019 *School Nutrition and Meal Cost Study*, 83% of schools incorporate food and nutrition activities into curriculum, but roughly two-thirds of schools that require food and nutrition education provide 10 or fewer hours of instruction per year [11]. Studies suggest that students typically receive between 4.5 and 13 h [12,13]. To expand their capacity, schools can partner with food and nutrition education programs (FNPs) run by nonprofits, hospitals, companies, government agencies, and universities [14].

Studies show that partnerships with community players like those that operate FNPs can bring resources into schools to help meet students′ needs and increase opportunities ′′for meaningful, pro-social engagement′′ [15]. Community involvement may also help ensure that policies, programs, and practices reflect the community′s culture. Yet national data show that a low percentage of school employees currently work with community partners to improve the school food environment. For example, the 2019 *School Nutrition and Meal Cost Study* reports that only about a quarter of school food authorities include community organizations when planning school meals, share information with a nutrition advisory council, or meet with community members to plan and assess nutrition education and promotion activities [16]. The same study shows that only 50.6% of schools address and implement food and nutrition education in their local wellness policies, 17.2% participate in a farm to school program, and 7.4% operate a school garden.

To increase a school′s capacity, FNPs can provide teacher professional development, access to staff with food and nutrition expertise, and resources such as curricula and gardening supplies. Some FNPs involve field trips to farmers markets or botanical gardens, while others improve school environments by adding gardens and kitchens where staff can teach. Others still may provide experiential opportunities where students reinforce academic standards for subjects like Science and Social Studies.

By providing additional resources and experiences, FNPs can help schools fulfill federal, state, and city requirements designed to integrate health in the classroom, cafeteria, and playground. For example, FNPs can help schools comply with the United States Department of Agriculture′s local wellness policy mandate to provide nutrition education. They may also maximize participation in the National School Lunch Program. Engaging students in the cafeteria, classroom, and school garden can help to ensure that students eat the meals that schools serve [17].

FNPs′ role in helping schools meet federal, state, and city requirements became increasingly important in the decade leading up to this study when U.S. policymakers took significant steps to support healthier school food environments. In 2010, First Lady Michelle Obama launched the ′′Let′s Move!′′ campaign to address childhood obesity, and President Obama created the first-ever Taskforce on Childhood Obesity [18]. The goals of the Taskforce were to review federal programs and policies relating to child nutrition and physical activity and develop a national action plan. The same year, Congress also passed the Healthy, Hunger-Free Kids Act (HHFKA) to improve the dietary quality of school meals; promote student wellness through a variety of means including food and nutrition education; and incorporate policy, systems, and environmental changes into SNAP-Ed, the largest nutrition education program in the U.S. [19]. As schools began implementing components of HHFKA, the U.S. Centers for Disease Control and Prevention launched their updated school health framework—the Whole School, Whole Community, Whole Child (WSCC) model—to help schools integrate health across academic subjects and spaces [20].

New York State reorganized its SNAP-Ed program, placing greater responsibility in community-based organizations (CBOs). In New York City (NYC), the Bloomberg administration introduced sweeping policy and programmatic changes to the school food environment, starting in 2007 with the introduction of water jets into schools [21]. In the years that followed, NYC policymakers created a farm to school program, Garden to Café (2008); adopted city-wide standards for the foods City agencies served, including school food (2008); launched a school gardens program, Grow to Learn (2010); created an Obesity Taskforce (2011); unveiled the annual City Food Metrics report to track the number of meals, number of schools gardens, kinds of food and nutrition education that the City supported (2011); and introduced salad bars into school cafeterias (2013).

Policies and programs, like the ones ushered in the decade prior to this study, can help to improve student health [16]. Effective policies and programs are informed by data, but comprehensive data on the organizations that run FNPs, their program characteristics, or distribution across schools did not exist in NYC. This study aims to help local education and health agencies assess characteristics of food and nutrition education in schools, as well as measure progress implementing school policies and practices. With 1840 schools and 1.1 million students, NYC, the nation′s largest school district [22], could serve as a model for other large urban school districts to assess FNPs across their schools. Ultimately, policymakers, advocates, funders, and schools need complete information to further integrate food and nutrition education across the classroom, cafeteria, and playground.

## 2. Materials and Methods

A cross-sectional study to determine the landscape of FNPs across the 1840 NYC public schools was conducted during the 2016–2017 school year. Quantitative and qualitative survey data on organizations and the FNPs they operate were collected. Data about the schools in which FNPs operate were gathered. To determine the distribution of FNPs across schools, a publicly available school data from the NYC DOE website was used, New York State School Report Card, and Accountability Reports.

### 2.1. Survey Development and Dissemination

An initial list of 65 organizations that operate FNPs was developed based on previous contacts. ′′Snowball sampling′′ helped expand the sample. Seventy-two different organizations implementing 180 FNPs in NYC schools during the 2016–2017 school year were identified. All identified organizations were invited to a project launch meeting in June 2016; over 30 representatives attended. Attendees identified topics to address in an FNP survey. Using a similar, previously-developed survey tool [14], feedback from the launch meeting was used to design a new draft survey. Five representatives from organizations that operate FNPs served on a committee to develop the survey, and 11 representatives from organizations piloted the survey, which led to refinements to the draft survey, resulting in a final survey. The final survey included 45 items on organizations and the FNPs they operate, including inputs, outputs, and outcomes. The final survey was conducted using Qualtrics©.

To encourage organizations to complete the survey, all contacts were emailed multiple times during a designated outreach period. Organizations that did not submit by deadline were contacted again, this time by phone. Given many contacts′ limited capacity, offers were made to help initially non-responsive organizations clean raw data or to fill out the survey along with the organization′s employees.

### 2.2. Creating an FNP Database

Based on a survey completed in October 2016, program-specific data for each organization and FNP were incorporated into the database, using Microsoft Excel for database construction and management. These data included information on FNP activities, occurrence, audience, school type, geographic location, academic subject areas, lesson content, and language.

### 2.3. Creating a School Database

A list of the 1840 public schools that NYC DOE operated was downloaded as a Microsoft Excel file from the Department′s website [22]. This list included school name and grade level information. To this list, data on school location, demographics, and student achievement from the New York State School Report Card and Accountability Reports were added [23]. Based on the address of each school, information on the 59 NYC Council Districts was also added.

### 2.4. Creating an FNPs in Schools Database

In June 2017, organizations that operate FNPs were asked to identify the schools in which they implemented FNPs during the 2016–2017 school year. A column for each FNP was added to the school database, to determine which schools during the 2016–2017 school year had which FNPs and how many each school had. Database set up and data collection methods were modeled after a 2011–2012 study of a subset of schools, elementary schools in the NYC boroughs of Brooklyn, Queens, and Manhattan. The Bronx and Staten Island were not included in this dataset.

### 2.5. Data Analysis

Several descriptive analyses were conducted on the databases using IBM SPSS Statistics Version 23 for Mac. The organization database was analyzed to identify trends across different entities that run FNPs. The FNP database was analyzed to determine characteristics for the FNPs operating in New York City schools. The schools database was analyzed to understand the geographic reach and relative size of FNPs, as well as characteristics of schools that partner with FNPs, such as student population demographics. FNPs reported the schools where they conducted programing, creating a continuous variable (range: 1 to 627 schools). From the list of schools where programming occurred, five size groupings of FNPs were created using natural breaks in the data: very small, small, medium, large, and very large. Programs that were funded by the federal government′s flagship nutrition education program, SNAP-Ed, were also labeled.

A final analysis was conducted to compare the 2011–2012 data from elementary schools in three of the five New York City boroughs to 2016–2017 data from elementary schools in the same three boroughs. The goal of this analysis was to understand if there were changes in the number of schools that partnered with FNPs in these five years.

## 3. Results

### 3.1. Characteristics of Organizations that Operate Food and Nutrition Education Programs in New York City Schools

Of the 72 organizations identified, 40 provided survey data on organizational characteristics. Eighteen of the 32 that did not complete the survey declared that their organizations no longer provided food and nutrition education; 14 others did not respond despite extensive outreach. Based on the research team′s decades of working with food and nutrition education organizations in NYC schools, the reach of the non-responsive organizations was determined de minimis.

Approximately 73% of respondents were nonprofits. See Table 1. For-profit entities constituted the next largest group of food and nutrition education providers (15%). Other common food and nutrition education providers included institutes of higher education, government agencies, and hospitals. Fifty-five percent of organizations had five or fewer fulltime employees conducting food and nutrition education, while 10% had more than 11.

The number of employees providing food and nutrition education may be a product of funding dedicated to FNPs. Approximately 33% of organizations had total annual budgets of less than USD 500,000, and 28% of organizations reported spending less than USD 250,000 on food and nutrition education, meaning for many organizations, there were limited funds to cover staff salaries. However, 10% of organizations spent more than USD 1 million annually on food and nutrition education programming. Across organizations of all sizes, funding was a top concern. Respondents categorized limited sources of funding, too narrowly focused funding, and lack of capacity to apply for funding as barriers to program expansion.

Notably, 32% and 37% of respondents did not provide data on the organizations′ total budget, or the amount of that budget dedicated specifically to food and nutrition education, respectively. This is may be because the survey respondents, many of whom were educators rather than budget administrators, did not have access to complete funding and budget information.

### 3.2. Administrative and Funding Characteristics of Food and Nutrition Education Programs in New York City Schools

The forty responding organizations operated 101 FNPs in NYC public schools. Many of the FNPs (43%) formed after 2010, and 73% operated exclusively in NYC. See Table 2. To support these programs, FNPs relied on a wide variety of funding sources. Together, government grants and contracts—city, state, and federal—were the most common sources of funding for individual programs (31%). Foundations and fee-for-service were the next most common, at 17% each. Private donors, company gifts, and fundraising events were other common sources.

More than half of FNPs operated in the Bronx, Brooklyn, and Manhattan, boroughs that are serviced by the most frequented subway lines [24]. Fewer operated in Queens and Staten Island, boroughs that are less densely populated and comparatively less accessible by public transportation [25]. For example, while 59% of programs served Bronx students, only 12% of programs served students in Staten Island.

FNPs also varied by the number of students they reached. Eighteen percent reached fewer than 100 students, and an equal percentage (18%) reached more than 2000 students. Between 101 and 500 students was the most common range of students reached. Data on the number of schools FNPs serve helps to provide a more complete picture of programs′ reach, suggesting that many of the programs that serve between 101 and 500 students may concentrate on a limited number of schools. Nearly two-thirds (64%) of FNPs are ′′very small′′ or ′′small′′, meaning they serve 10 or fewer schools. Of FNP programs, very small and small programs comprise 10% of total school reach. In contrast, only 5% of FNPs are ′′very large′′, reaching more than 127 schools. Yet these very large FNPs makeup 55% of total school reach.

The five ′′very large′′ programs receive significant government support (see Table 3). Two programs, CookShop (SNAP-Ed) and the Expanded Food and Nutrition Education Program (EFNEP), are funded through the U.S. Department of Agriculture′s largest nutrition education programs. Two others, Grow to Learn and Garden to Café, the Bloomberg administration created and government or quasi-government agencies continue to operate. (Technically a non-profit, GrowNYC, operates out of city-owned offices and in City-sanctioned public spaces.) For the last, City Growers’ School Gardens Workshops, government funding was the largest single source of the organization′s income. The federally-funded programs, CookShop and EFNEP, both serve low-income populations, whereas the locally-supported FNPs serve the general student population, 75% of which is low-income. The federal programs and Garden to Café focus on cooking, healthy eating, and school meals, covering many of the nutrition and dietary behavior topics that CDC recommends. Grow to Learn, School Gardens Workshops, and Garden to Café support school gardens, reaching at least one-third of NYC public schools. These three programs alone support gardens in a greater percentage of NYC schools than occurs nationally; according to the Department of Agriculture′s 2019 *School Nutrition and Meal Cost Study*, only 7.4% of schools across the country operate a school garden.

Notably, only one of the ′′very large′′ programs, CookShop, was SNAP-Ed funded. SNAP-Ed is the largest single source of funding for nutrition education in the City, bringing in more than USD 5 million a year. These organizations reached between 11 and 208 schools, accounting for 15% of total FNP school reach. See Table 4. The organization that provided CookShop, Food Bank For New York City, reached nearly three times the number of schools as the next largest SNAP-Ed provider and operated in all five boroughs.

### 3.3. Service Attributes of Food and Nutrition Education Programs in New York City Schools

FNPs varied by goals, content, activities, and physical location, as shown in Table 5. Common FNP goals included changing participants′ behaviors, as well as improving attitudes, knowledge, awareness, and skills. When FNPs responded about the targets for their programming and which of these targets were measured as evaluation outcomes, ′′improved knowledge and awareness′′ was the most common target and also the most commonly evaluated target. Pre- and post-program surveys were the most common form of evaluation.

A majority of FNP curricula covered nutritional knowledge and recipes. Over 40% focused on ecology, and nearly half focused on food justice, environment, and access. Only 21% focused on media literacy and 31% on diet-related diseases. Nearly 70% of FNPs offered lessons to address science learning objectives. Literacy and math were also common subjects.

Popular programming activities included cooking (70%), classroom lessons (67%), gardening (46%), and field trips (31%). Most FNP activities were designed for students, but about half (49%) of programs included activities that targeted and operated activities for families.

Program staff were the most common instructors for 59% of FNPS; school teachers taught 18% of FNPs, demonstrating how FNPs build schools′ capacity. Approximately 24% of FNPs occurred only on school property, such as in a classroom or school garden. Thirty-seven percent of programs combined school-based with off-site learning, for example, reinforcing classroom-based learning by taking students to farmers markets. Approximately 34% of FNPs occurred only offsite, for example, as a fieldtrips or workshops at a botanical garden.

Only 24% of FNPs were offered in Spanish. No other languages had significant offerings, even though NYC students′ families speak over 180 languages and NYC schools produce materials in the nine other official City languages, besides English.

### 3.4. Food and Nutrition Education Program Distribution in New York City Schools

In the 2016–2017 school year, 56% of NYC public schools, or 1025 schools, partnered with at least one FNP. Eight hundred and fifteen schools (44%) had no partnership with an FNP (see Table 6). The percentage of schools that partnered with FNPs varied by school location, type, poverty, and race.

Manhattan and Brooklyn had the highest rates of schools partnering with FNPs; approximately 58% of schools in those boroughs worked with FNPs. The Bronx and Queens had slightly lower than average rates: roughly 55%. Staten Island had the lowest rate, with 43%, or 34 of the borough′s 80 schools, partnering with an FNP.

When it came to students′ age, schools with lower grade-levels were more likely to partner with FNPs. Elementary schools had the highest rate of FNPs (70%) and high schools the lowest. Only one-third of high schools partnered with one or more FNPs, and very few partnered with more than three.

For poverty, where the percentage of students eligible for free or reduced-price lunch was used as a proxy, schools with the lowest and highest rates were more likely to partner with FNPs. These schools had higher percentages of schools with FNPs than the citywide average of 56%. Seventy-six percent of schools with fewer than 10%, 86% of schools with 10.1–20%, and 68% of schools with 20.1–30% of students living in poverty partnered with FNPs; but these schools with less than 30% of students in poverty represent only 5% of total public schools in NYC. In contrast, schools with more than 90% of students who live in poverty make up more than one-third of public schools. A high percentage (63%), but not as high as in wealthy schools, partnered with an FNP.

The percentage of schools that partnered FNPs also varied by race. NYC′s public schools are diverse, but segregated. Approximately 41% of students are Latinx, 26% are Black, 16% are Asian, and 15% are White [26]. The quintile of schools where students were most likely to match the system′s demographics as a whole, Quintile 2, had the highest rate of partnership with FNPs (60%). In contrast, the quintile of schools with the highest proportion of Latinx and/or Black students (Quintile 5, 96.3–100%) had the lowest rate of partnership with FNPs (53%). While the overall rates were the lowest for this group, further analysis, not shown in tables, reveals that these schools were more likely to partner with an FNP that incorporated media literacy. City-wide, only 5% of schools with FNPs have programs that address media literacy, but in schools where more than 96% of the student population was Latinx and/or Black, 20% partnered with an FNP that addressed media literacy, suggesting a responsiveness to community needs.

Further analysis also revealed that FNPs may not be as responsive to a community′s language needs. Of the 732 schools with a majority Latinx population, 429 partnered with at least one FNP. Only half of these schools, or 215, partnered with an FNP that translated materials into Spanish.

### 3.5. Changes to Food and Nutrition Education Program Distribution between 2011–2012 and 2016–2017

To understand if the percentage of schools partnering with FNPs changed over time, data from 2016–2017 were compared to data from a 2011–2012 prior study [14]. The 2011–2012 data were for elementary schools in Brooklyn, Manhattan and Queens. For this subset of NYC public schools, there was an increase, with a greater percentage of schools partnering with FNPs in 2016–2017 (see Table 7).

## 4. Discussion

FNPs can play an important part in ensuring that all public school students have access to great food and nutrition education. This research suggests that, while more NYC public schools have partnered with FNPs, access across schools is not equitable.

This study′s results highlight the importance of government funding and policies. Government grants and contracts were the most common sources of funding for FNPs (Table 2). Public policies and public funding were also intertwined with the largest programs′ school reach. Approximately 73% of organizations providing food and nutrition education were nonprofits (Table 1), and government agencies represented less than 3% of providers. Yet, of the five organizations that operated the ′′very large′′ FNPs, three were government or quasi-government agencies (Table 3). The other two organizations operating very large FNPs received all or a majority of their funding from the governments, suggesting that public funding and policies may be important to scaling programs.

Notable increases in FNP reach occurred in Brooklyn, Manhattan, and Queens elementary schools between the 2011–2012 and 2016–2017 (Table 7), a period following a flurry of policy activity. The increase in schools that partnered with FNPs was potentially driven by significant growth from the City′s Grow to Learn program. The program launched in 2010, and by 2016 was the City′s largest FNP (Table 3), thanks, in part, to the Mayor′s policy priorities [27]. In fact, the largest percentage of FNPs started after 2010, in the wake of obesity initiatives at the federal and local levels. In 2010, ′′Let′s Move′′ and HHFKA put a spotlight on the need to improve school food environments, building on local successes from policymakers like Bloomberg.

Government funding and policies can work in tandem to support nutrition education best practices. It is important that food and nutrition education connect to students’ lives, values, and interests [1,2], so it is encouraging that so many of the FNPs focus on experiential learning through cooking, gardening, and fieldtrips (Table 5). Experiential learning can be resource-intensive, so financial support may be necessary, and few funders beyond the government have access to capital at the scaled needed in NYC. Policy, systems, and environmental changes can be low cost activities to improve the school food environment, and legislation can help encourage these nutrition education activities. For example, HHFKA incorporated policy, systems, and environmental changes into SNAP-Ed programming [3].

Finally, government funding can help to correct systemic inequities, and policies can help to ensure these inequities are not replicated in the future. This study demonstrated that inequities exist across race, language, geographic, and income divides. For example, FNPs were less likely to partner with schools with the highest rates of Black and/or Latinx students (Table 6). Yet Black and Latinx children and adolescents are at greater risk for diet-related diseases like obesity [28]. Limited food and nutrition education resources should be concentrating where need is highest, including in schools where there are higher rates of students prone to diet-related diseases. Government policies can help ensure that funds are directed to schools with predominantly Black and Latinx populations; there are numerous examples of designated at the federal level, such as the Racial and Ethnic Approaches to Community Health (REACH) grant.

Another way that policy can help ensure FNPs are more responsive to communities′ needs is by supporting translation services for FNP materials. For example, two-thirds of ELL students in NYC schools speak Spanish at home [29], yet only 29% of schools with a majority Latinx population worked with an FNP that provided materials in Spanish (Table 6). NYC already translates a host of documents into the official City languages [30]; the City could go further, both requiring and providing additional funding for translation services for school partners. Policymakers can also make additional resources available for FNPs to address geographic disparities. For example, in Staten Island, where only 12% of FNPs operate (Table 2), the Borough President has invested heavily in a teaching farm, hosted meetings for new FNPs, and led school tours to introduce FNPs to school principals.

Finally, legislation can also help to address disparities in family income. At the federal level, the U.S. Department of Agriculture has set income specifications in regulation to ensure SNAP-Ed and EFNEP are available to those who cannot otherwise afford these programs. In 2019, New York City passed a law mandating that the Department of Education publish data on schools′ Parent Teacher Association (PTA) data. PTA fundraising can create unequal realities, where wealthy schools can raise money to pay for extra-curriculars like FNPs, while schools with lower-income families cannot. Analysis by a local newspaper found that the median white student attends a school that raises 16 times as much as the median black student [31], suggesting that PTA funding could influence FNP distribution and reach in schools with high proportions of Black students. A next step could be to pass legislation requiring a fund-sharing system for PTAs across the City. Ultimately, limited food and nutrition education resources should be concentrating where need is highest, and government funding and policies can be key components to addressing inequities.

### Limitations and Future Research

This study has several limitations. First, for many of the survey questions there was a high percentage of missing data, despite efforts to contact and assist the organizations. The missing data indicate the challenges to better understanding and supporting FNPs in schools, given CBOs′ limited capacity. Future research on FNPs in schools may consider providing financial incentives to increase organizations′ capacity to provide data. Second, because this was a landscape assessment and the first full study of FNPs across NYC, it is only a descriptive analysis. Future research could track for statistically significant changes in FNP characteristics and reach over time. Third, this study considered the reach of FNPs by school. Not all FNPs reach all students in a school. Typically, FNPs do not provide education to all students in the schools where they work. If the goal of research is that all the K-12 students are reached by FNPs, more detailed reporting is needed. Ideally, school districts would do this reporting, tracking what grades FNPs reached. Finally, this study was not able to categorize FNPs by intensity of the education provided. FNPs vary from one-time field trips to multiple lessons in a week throughout the school year. The intensity of FNPs could be investigated in future research.

To address service gaps in New York, future research could examine which FNP characteristics make it less likely that FNPs serve minority and English language learners. Researchers could analyze information on activities, content, academic subjects addressed by school level (elementary, middle, high) to provide insight on how to increase food and nutrition education programming in high schools. Alternatively, future studies could look at how schools partnered with and integrated multiple FNPs to encourage further collaboration.

Other cities could replicate portions of this study to determine the degree to which organizational characteristics, program characteristics, school make-up, and government policies and funding are related to FNP reach. Studies in other jurisdictions could help determine to what degree findings from this paper reflect or do not reflect patterns in large urban school districts.

## 5. Conclusions

To ensure equitable access, more coordination, investment, and collaboration are needed. Foundations and officials can enhance FNP capacity by providing funding for technical assistance, tools, and training. School administrators, teachers, school food service, parents, and students can continue to work together to strengthen school-based food and nutrition education. Policymakers can support laws and funding measures that provide FNPs with the sort of stability needed to grow programs′ reach. Since this research was conducted, organizations that operate FNPs have taken meaningful steps to create a network that coordinates food and nutrition education across NYC schools, advocates for policies to support food and nutrition education, aligns evaluation strategies, and bolsters efficiencies through shared resources. Organizations across the U.S. can use the methods described here to identify service gaps and strengthen food and nutrition education in their own cities and towns.

## Figures and Tables

**Table 1 nutrients-12-02423-t001:** Characteristics of Organizations Operating Food and Nutrition Education Programs in New York City Schools (*n* = 40 organizations).

Characteristic	Percentage of Organizations
Organization Type	
Nonprofit	73%
For-profit	15%
Other ^a^	12%
Organization total budget	
≤USD 500,000	33%
>500,000	35%
did not provide data	32%
Organization budget toward food and nutrition education	
≤USD 250,000	28%
USD 250,000–500,000	15%
USD 500,000–USD 1,000,000	10%
>USD 1,000,000	10%
did not provide data	37%
Barriers to increase or sustain funding (check all that apply) (34 organizations provided data)	
funding term too short	64%
lack of capacity to apply for grants & funding	68%
funding only supports one aspect of programming	84%
limited funding pool available	88%
Fulltime employees conducting food and nutrition education	
0–5	55%
6–10	13%
11+	10%
did not provide data	22%

^a^ Higher education and government agencies each represent 2.5% of organizations that chose ′′Other′′.

**Table 2 nutrients-12-02423-t002:** Administrative and Funding Characteristics of Food and Nutrition Education Programs (FNPs) (*n* = 101 programs).

Characteristic	Percentageof Programs
FNP start year	
≤2000	9%
2001–2010	27%
2011–2017	43%
did not provide data	21%
FNP geographic scope	
national	7%
New York State	1%
New York City	73%
did not provide data	19%
FNP reach by NYC borough ^a^	
Bronx	59%
Brooklyn	53%
Manhattan	57%
Queens	43%
Staten Island	12%
FNP reach by number of students	
1–100 students	18%
101–500 students	25%
501–2000 students	18%
> 2000 students	18%
did not provide data	21%
FNP size by number of schools reached ^b^	
Very small (1–3 schools)	33%
Small (4–10 schools)	31%
Medium (11–19 schools)	19%
Large (30–88 schools)	12%
Very Large (127–627 schools)	5%
FNP size as a proportion of total FNP reach ^c^	
Very small (1–3 schools)	2%
Small (4–10 schools)	8%
Medium (11–19 schools)	11%
Large (30–88 schools)	23%
Very Large (127–627 schools)	55%
FNP funding source (check all that apply) (58 programs provided data)	
City grants and/or contracts	10%
State grants and/or contracts	5%
Federal grants and/or contracts	16%
Foundations	17%
Companies	6%
Fundraising events	5%
Private donors ^d^	9%
Program fees	17%
Other	13%

^a^ ′′Reach by borough′′ indicates programs that partner with at least one school in the geography specified. ^b^ FNP size is a continuous variable as FNPs reported a list of schools in which they worked. There were no FNPs that reported working in 20 to 29 schools or in 89 to 126 schools. ^c^ ′′Total FNP reach′′ represents the total number of FNPs across all 5 boroughs. ^d^ ′′Private donors′′ was the most common response written in ′′Other′′. Since this option was pulled from ′′other′′ to create this category of funding, additional FNPs may have also received funding from private donors.

**Table 3 nutrients-12-02423-t003:** Snapshot of the ′′Very Large′′ Food and Nutrition Education Programs.

FNP Name	Organization Name	Organization Type	Number of Schools Reached	Government Support for FNP	FNP Focus
Grow to Learn	GrowNYC	quasi-governmental ^a^	627	city	gardening, environmental education
CookShop (SNAP-Ed)	Food Bank For New York City	non-profit	173	federal	cooking, healthy eating
School Garden Workshops	City Growers	non-profit	172	city	gardening, environmental education
Expanded Food and Nutrition Education Program (EFNEP)	Cornell Cooperative Extension NYC	quasi-governmental ^a^	131	federal	cooking, healthy eating
Garden to Cafe	Department of Education Office of Food & Nutrition Services	government	127	city	school meals, healthy eating

^a^ ′′Quasi-governmental′′ refers to non-profits that have very close ties politically and financially with government agencies. Both GrowNYC and Cornell Cooperative Extension NYC are technically independent non-profits, but the exist largely to operate government programs.

**Table 4 nutrients-12-02423-t004:** SNAP-Ed-Funded Food and Nutrition Education Programs Funded in NYC Schools during the 2016–17 School Year.

Organization Name	Number of SNAP-Ed Funded FNPs	Number of Schools Reached	Percentage of Total FNP School Reach	Boroughs FNPs Served
Children′s Aid	6	70	3%	Bronx, Manhattan
City Harvest	6	36	2%	Brooklyn, Queens, Staten Island
Food Bank For New York City	2	208	9%	Bronx, Brooklyn, Manhattan, Queens, Staten Island
NY Common Pantry	2	11	1%	Bronx, Manhattan
TOTAL	16	325	15%	Bronx, Brooklyn, Manhattan, Queens, Staten Island

**Table 5 nutrients-12-02423-t005:** Service Attributes of Food and Nutrition Education Programs (FNPs) (*n* = 101 programs).

Service Attribute	Percentage of Programs
FNP session length	
<1 h	22%
1–2 h	29%
2–4 h	13%
4+ hours	11%
did not provide data	25%
FNP targets for programming and outcomes measured in evaluation (check all that apply) (78 programs provided data) ^a^	
Change behavior	
FNP targeted *changing behavior* in programming	69%
FNP measured the outcome of *changing behavior*	40%
Change attitudes	
FNP targeted *changing attitudes* in programming	62%
FNP measured the outcome of *changing attitudes*	39%
Improve knowledge and awareness	
FNP targeted *changing knowledge and awareness* in programming	72%
FNP measured the outcome of *changing knowledge and awareness*	48%
Improve skills	
FNP targeted improving *skills* in programming	62%
FNP measured the outcome of *improving skills*	30%
Change environment	
FNP targeted changing *environment* in programming	33%
FNP measured the outcome of *changing environment*	14%
FNP activities ^b^ (check all that apply) (80 programs provided data) ^a^	
Cooking (students)	70%
Classroom lessons (students)	67%
Family involvement and activities (families)	49%
Gardening/farming (students)	46%
Fieldtrips (students)	31%
Professional development (teachers)	24%
Food environment change (environment)	20%
FNP curriculum content areas (check all that apply) (88 programs provided data) ^a^	
Nutrition knowledge	94%
Recipes	91%
Growing food & gardening skills	64%
Food culture	63%
Family meals	52%
Food environment & access	51%
Food safety	51%
Food justice	49%
Ecology	42%
Obesity and other diet related diseases	31%
Human body systems	25%
Media literacy	21%
Eating disorders	6%
Other	33%
FNP academic subjects addressed ^c^ (check all that apply) (76 programs provided data) ^a^	
Science	69%
Literacy	59%
Math	55%
Social studies	37%
Arts	30%
FNP available in other languages beside English^d^ (check all that apply) (80 programs provided data) ^a^	
Spanish	24%
Chinese	3%
FNP implementer	
School teachers (alone or with others, e.g., program staff or volunteers)	18%
Program staff (alone or with others, excluding teachers)	59%
Volunteers, interns, or other	3%
Did not provide data	20%
FNP locations for programming	
Always in schools	24%
Sometimes in schools, sometimes in other settings	37%
Always in other settings	34%
Did not provide data	5%

^a^ Percentages are for all programs (*n* = 101); program that did not provide data counted as ′′no′′. ^b^ Only activities >20% of programs are reported here. Student leadership training (16%) and student wellness policy/councils (13%) were other activities. ^c^ 11% of programs chose ′′Other′′ and 3% chose ′′None′′ ^d^ New York City public school students families speak more than 180 languages. One percent of programs checked ′′All other official NYC languages′′ (Arabic, Bengali, French, Haitian Creole, Korean, Russian, Urdu). Four percent of programs checked ′′Other′′.

**Table 6 nutrients-12-02423-t006:** Characteristics of Schools Partnering with Food and Nutrition Education Programs during the 2016–2017 School Year (*n* = 1840 schools, *n* = 101 programs).

Characteristic	Percentage of Schools
Number of FNPs	
0 FNPs	44%
1 FNP	28%
2 FNP	14%
3 FNP	7%
4–5 FNPs	5%
6+ FNPs	2%
Borough	
Bronx (*n* = 447 schools)	55%
Brooklyn (*n* = 574 schools)	58%
Manhattan (*n* = 360 schools)	58%
Queens (*n* = 379 schools)	55%
Staten Island (*n* = 80 schools)	43%
School type	
Elementary (*n* = 734 schools)	69%
Elementary-middle (*n* = 218 schools)	67%
Elementary-middle-high (*n* = 64 schools)	64%
Middle only (*n* = 289 schools)	50%
Middle-high (*n* = 109 schools)	46%
High school (*n* = 426 schools)	32%
Poverty rate ^a^	
0–10.1% of students in poverty (*n* = 25 schools)	76%
10.1–20% of students in poverty (*n* = 22 schools)	86%
20.1–30% of students in poverty (*n* = 37 schools)	68%
30.1–40% of students in poverty (*n* = 64 schools)	50%
40.1–50% of students in poverty (*n* = 74 schools)	51%
50.1–60% of students in poverty (*n* = 93 schools)	58%
60.1–70% of students in poverty (*n* = 151 schools)	52%
70.1–80% of students in poverty (*n* = 360 schools)	48%
80.1–90% of students in poverty (*n* = 394 schools)	49%
90.1–100% of students in poverty (*n* = 620 schools)	63%
Quintile of students who are Black and/or Latinx ^b^	
Quintile 1: 2.3–46.9%	58%
Quintile 2: 47.0–81.7%	60%
Quintile 3: 81.8–91.0%	54%
Quintile 4: 91.1–96.2%	54%
Quintile 5: 96.3–100%	53%

^a^ Percentage of students who qualify for free- or reduced-price lunch was used to represent students in poverty. ^b^ Data are presented in quintiles, *n* = 368 per quintile.

**Table 7 nutrients-12-02423-t007:** Comparison of 2011–2012 and 2016–2017 FNP Data for Elementary Schools in Three Boroughs.

Brooklyn, Manhattan, and Queens Elementary Schools Partnering with FNPs	2011–2012 School Year	2016–2017 School Year
0 FNPs	61%	29%
1 FNPs	25%	32%
2 FNPs	8%	17%
3+ FNPs	6%	22%

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
