# Peer review of "Expanding and Enhancing Food and Nutrition Education in New York City Public Schools: An Examination of Program Characteristics and Distribution"

_nutrients, 2020, doi:10.3390/nu12082423_

Round 1
Reviewer 1 Report
2.5 Data Analysis: The "2.5 Data Analysis" section is incomplete, given sections 2.1-2.2 above and the fact the purpose of this analysis is to be more comprehensive across NYC five boroughs (not only three) in 2016-2017 (vs. in 2011-12). We only know they used Qualtrics for final survey, and assumed Excel or Access (?) for databse construction and management. What were the secondary statistical analyses? What software used? Details!
3.1 Results/Table 1: There are several issues with Table 1. First, only 40 of 72 organizations provided "reliable survey data?" Do these 40 cross the five boroughs? We must know breakdown by borough in text or in this table as second set of rows! Second, in Table 1 (not done in Table 2!?), the footnote "c" is unnecessary if the text is in parens already regarding answer options for this question (right now, unnecessary redundancy) Third, several sets of rows have too much missing data...22-33%. The table must focus on the 40 of 72 organizations; alternatively, if among 40 organizations providing reliable survey data there were still missing data for certain questions, then in this reviewer's opinion this study is not publishable and it is simply a report for local use in NYC.
3.2 Results/Table 2: There are several issues with Table 2. First, only 40 of 72 organizations provided "reliable survey data" on 101 FNPs, with 43% after last study period and 36% before plus ~2-in-10 no data/unknown again (see comments at Table 1). Do these 101 cross the five boroughs? We must know breakdown by borough in text or in this table as second set of rows! Second, again, several sets of rows have too much missing data...20 +/- 1%. What does this mean, among the 101 still are missing data for key variables? Again, if among 101 FNPs from 40 organizations providing reliable survey data there were still missing data for certain questions, then in this reviewer's opinion this study is not publishable and it is simply a report for local use in NYC. Third, table must be split into two, there is too much going on. A separate table should be created about service attributes, vs. admin./$.
3.1-2 Results/Table 3: Table 3 now Table 4 (see comments at Table 2 above). This part of Table 3 highlighted is important; we need something similar for entire study period at/with Table 1 or revised Table 2.
- Discussion:
Please refer back to your tables (in parens) with data-specific statements in paragraphs within lines 208-215.
There is nothing in this Discussion of generalizable value outside NYC, especially given "missing data" issues above. Discussion must be expanded to other similar urban/metropolitan areas of USA, otherwise, as noted above, not publishable in peer-review journal (only a local report).
There are four paragraphs in Discussion between lines 216 and 238. Instead, there should be two paragraphs (lines 216-229 and 230-238), given similar topics covered.
Reviewer 2 Report
The topic has interest to those working in nutrition education and child nutrition. The information on FNPs may be more useful if revised to provide more insight into reasons for FNP operational decisions. It would also be useful to provide published information on nutrition education in schools nationwide to place findings in a larger context.
Specific comments are:
Introduction: Provide some context by using national data such as from USDA's School Nutrition and Meal Cost Study and/or Child Nutrition Program Operations Study or CDC's School Health Policies and Practices Study to give available national data on amount of nutrition education being provided in schools, the extent to which it ise being provided by community groups; any relevant background such as share of schools with farm-to-school programs, community gardens, etc.
Results, Section 3.1: I'd like to know a little more about how FNP organizational characteristics were associated with the schools in which they delivered services and the number of schools in which they delivered services. Specifically:
a) What kinds of criteria do FNPs have for where they deliver programming? For example, SNAP-Ed is a major source of nutrition education funding, but can only operate in schools that meet income criteria (I believe 50% or more free/reduced price meals). Please identify SNAP-Ed and other programs that have income or need-based criteria, and other selection criteria that may determine where FNPs do programming. For example, are some part of farm-to-school or school garden programs?
b) Lines 122-126: It seems most FNPs are small organizations, but 10% have considerably more funds. Presumably these big-budget FNPs offer more programming, in more schools. Please provide some information on these large programs, the number of schools they serve, their mandate and educational priorities (for example, for SNAP-Ed, it would be to serve low-income children), and how this influences the schools where programming is offered and the type of programming offered. Maybe break down FNPs by income group and the number of schools served by low-, medium- and high-budget groups. It would be interesting to see to what extent the outside nutrition education in schools is being provided by just a few big-budget FNPs.
Table 2 -- please break out information on activities, content, academic subjects addressed by school level (elementary, middle, high). This could add insights into how to provide more FNP programming in high schools.
Table 3 -- Although breaking income data into quintiles may seem logical, if a major programmer uses another income break (e.g. SNAP-ED, 50% or more in poverty), your approach could obscure a major reason for differences. Consider presenting in a way that makes sense relative to program regulations for major FNP providers either in place, or in addition, to quintiles.
It might be useful to add a cross tab to Table 3 that looks at funding source for FNPs in schools of differing income level. Public programs like SNAP-Ed may be concentrated in low-income schools that meet their targeting criteria, whereas privately-funded programs may be located in well-off schools with parents with resources to seek out enrichment programs. This would result in the "skinny middle" of mid-income schools with fewer programs that is seen in your data.
Discussion
Line 206: "more NYC public schools..." -- do you mean "most" or "more compared to a past date?"--this is confusing. Can you find national data (maybe in a USDA or CDC source) to compare the level of FNP involvement in NYC schools to the national norm?
Lines 208-211: Besides size and scope, consider FNP mission and any criteria for involvement in schools--a focus on low income will lead an FNP to one set of schools; another focus may also play a role--for example, a focus on school gardens will lead to a school with interest and resources to support a garden.
Line 215: Re: lack of FNP work with high schools, does anything in the data provide insight as to why? Perhaps the breakout of content area by school level I suggested above would provide an opportunity to consider why this might occur and strategies for addressing it. Some content may fit better in a high school curriculum--those content areas could be encouraged.
Lines 216-229: Is there anything in the characteristics of the FNPs that would make it less likely that they would serve minority and ELL students? If so, how can this be addressed? Perhaps they need funding for language translations? The suggestion re: media literacy is interesting; more investigation of interests and needs of these groups could be helpful in not only providing more nutrition education but making it more relevant to their needs.
Reviewer 3 Report
Thank you for the opportunity to read this well-written article. This study provides a useful snapshot of the status of FNP in New York City. This would likely be of interest to stakeholders in education and health sectors in NYC and other large urban communities.
This paper could be strengthened through:
- a more fulsome discussion of the results and implications of results (e.g., how to address food and nutrition supports across all quintiles, characteristics of programs with greatest impact/reach).
- It would be helpful to describe examples of how schools partnered with and integrated multiple FNPs. This might make these results more useful for others who could benefit from such diverse partnership arrangements.
